# Identification of Male Sex-Related Genes Regulated by *SDHB* in *Macrobrachium nipponense* Based on Transcriptome Analysis after an RNAi Knockdown

**DOI:** 10.3390/ijms241713176

**Published:** 2023-08-24

**Authors:** Zijian Gao, Wenyi Zhang, Sufei Jiang, Huwei Yuan, Pengfei Cai, Shubo Jin, Hongtuo Fu

**Affiliations:** 1Wuxi Fisheries College, Nanjing Agricultural University, Wuxi 214081, China; gaozijiangenomics@163.com (Z.G.); yuan08102021@126.com (H.Y.); ckgg5436@126.com (P.C.); 2Key Laboratory of Freshwater Fisheries and Germplasm Resources Utilization, Ministry of Agriculture and Rural Affairs, Freshwater Fisheries Research Center, Chinese Academy of Fishery Sciences, Wuxi 214081, China; zhangwy@ffrc.cn (W.Z.); jiangsf@ffrc.cn (S.J.)

**Keywords:** crustaceans, *Macrobrachium nipponense*, RNAi, male sexual development, transcriptome analysis

## Abstract

The oriental river prawn (*Macrobrachium nipponense*) is a commercially important species in Asia. A previous study showed that the succinate dehydrogenase complex iron sulfur subunit B (*SDHB*) gene participates in testes development in this species through its effect on the expression of the insulin-like androgenic gland hormone gene. This study knocked-down the *Mn-SDHB* genes in *M. nipponense* using RNAi. A transcriptome analysis of the androgenic gland and testes was then performed to discover the male sex-related genes regulated by *SDHB* and investigate the mechanism of male sexual development in this species. More than 16,623 unigenes were discovered in each sample generated. In the androgenic gland, most of the differentially expressed genes were enriched in the hypertrophic cardiomyopathy pathway, while in the testes, they were enriched in the citrate cycle pathway. In addition, after *Mn-SDHB* knockdown, five genes were found to be downregulated in the androgenic gland in a series of biological processes associated with phosphorylated carbohydrate synthesis and transformations in the glycolysis/gluconeogenesis pathway. Moreover, a total of nine male sex-related genes were identified including Pro-resilin, insulin-like androgenic gland hormone, Protein mono-ADP-ribosyltransferase PAPR11, DNAJC2, C-type Lectin-1, Tyrosine-protein kinase Yes, Vigilin, and Sperm motility kinase Y-like, demonstrating the regulatory effects of *Mn-SDHB*, and providing a reference for the further study of the mechanisms of male development in *M. nipponense*.

## 1. Introduction

The oriental river prawn (*Macrobrachium nipponense*) is an economically important species in China, and is widely distributed in freshwater and low-salinity estuarine regions of China and other Asian countries [1,2,3]. The annual production of *M. nipponense* reached 225,321 tons in 2019 with a value of over 3 billion dollars, producing huge economic benefits [4]. A previous study showed that both the testes and ovaries of newly hatched *M. nipponense* can reach sexual maturity within 45 days after hatching [5]. This enables inbreeding between newborn shrimps, resulting in shrimps with a small market size and short life span, and therefore restricts the sustainable development of the *M. nipponense* industry. There is therefore an urgent need to understand the reproductive mechanisms of *M. nipponense*, in order to establish artificial techniques to regulate their gonad development processes.

Succinate dehydrogenase complex iron sulfur subunit B (*SDHB*) is a crucial protein subunit comprising one of the four components of succinate dehydrogenase, the only enzyme to be involved in the tricarboxylic acid cycle through its association with the inner mitochondrial membrane, where it catalyzes the oxidation of succinate [6,7]. Its particular iron–sulfur structure enables the succinate dehydrogenase complex to bind to two small integral membrane proteins of 13.5 and 15.5 kDa, which participate in the oxidative stress of oxidative phosphorylation, and play vital roles in the electron transport chain. The spermatogenic cells of *Lymnaea stagnalis* contain succinate dehydrogenase, which prevents metabolic acidosis in the testes through the anaerobic production of lactate and succinate by the Sertoli cells. They also play essential roles in developing rat testes [8]. *SDHB* also had multiple functions, including resistance to carboxin in *Ustilago maydis* [9], the prevention of superoxide generation and premature aging in *Caenorhabditis elegans* [10], and resistance to the fungicide boscalid in *Sclerotinia sclerotiorum* [11]. A previous study showed that knockdown of the expression of *SDHB* by RNAi in male *M. nipponense* can inhibit the development of the testes by affecting the expression of the insulin-like androgenic gland hormone (*IAG*) [12].

Many researchers have proposed that the eyestalk-androgenic gland-testis endocrine axis in male crustaceans plays an essential role in regulating male sexual differentiation and reproduction [13,14]. The androgenic gland and testes have therefore become the target tissues for studies of male differentiation and reproduction in crustacean species. The androgenic gland is a special organ in male crustaceans. The hormones it secretes play essential roles in testis formation and male secondary sexual characteristics [15,16,17]. *IAG* is specifically expressed in the androgenic gland, and has been shown to regulate the process of male sexual differentiation and reproduction in many crustaceans [18,19,20,21,22]. Sexual development and the regulatory mechanism of *IAG* in crustaceans are very complicated processes and rely on the expression of multiple genes. Recent studies have shown that several sex-related genes, including CFSH, Vtg, Wnt4, the Dmrt gene family, the Sox gene family, the cell cycle gene family, and Fem-1, participate in the regulation of *IAG*, either directly or indirectly. Nevertheless, further investigations are required to explore the interrelationships between sex-related genes and their association with *IAG* [23,24,25,26]. Knockdown of *IAG* using RNAi had a significant inhibitory effect on spermatogenesis in *M. rosenbergii* [27]. The testes play an essential regulatory role in reproduction, sexual maturity, and sex differentiation in *M. nipponense*. Some transcriptome analyses have been conducted on the testes of this species [28,29], and a number of genes selected from the testes have been shown to regulate its male reproductive processes [30,31,32].

In this study, we aimed to identify the genes regulated by *SDHB* in *M. nipponense* using transcriptome profiling analysis after *SDHB* knockdown using RNAi. The differentially expressed genes (DEGs) identified in this study may be involved in regulating male reproduction, providing valuable information regarding the mechanisms of male reproduction in crustacean species in general.

## 2. Results

### 2.1. Overview of the RNA-Seq of M. nipponense

Androgenic gland and testis samples were collected from both the control and the RNAi groups on day 7 after the injection of double-stranded RNA to form RNA-seq samples, namely SDHB_AG for the androgenic gland *SDHB* knockdown group; SDHB_T for the testis *SDHB* knockdown group; C_AG for the androgenic gland control group; and C_T for the testis control group. After quality control screening, about 84.34 Gb of the total raw reads (about 7.02 Gb for each sample) were generated. As the data show, the percentage of Q30 bases of all products exceeded 92.42%, indicating high quality sequences. Through alignment with the reference genome, the mapping rate exceeded 91.23% and the unigenes obtained exceeded 16,623 for each RNA-seq sample (Table 1).

All sequence reads were deposited in the Sequence Read Archive of the National Center for Biotechnology Information (NCBI) (accession SRR24335442-SRR24335459) under Bioproject PRJNA962267.

### 2.2. Identification and Functional Analysis of DEGs

Fragments per kilobase per million reads (FPKM) is a common measure for estimating gene expression levels in transcriptome sequencing data analysis. For the 12 samples in this study, the mean FPKM values for each gene ranged from 29.76 to 34.43, and the overall trends in gene expression were similar between the samples (Appendix A).

The DEG analysis results revealed a total of 67 DEGs in the “SDHB_T vs. C_T” comparison. Of these, 49 genes were upregulated and 18 were downregulated. Similarly, the “SDHB_AG vs. C_AG” comparison identified 235 DEGs, consisting of 52 upregulated genes and 183 downregulated genes.

### 2.3. GO and KEGG Enrichment Analysis of DEGs

The gene products of the total unigenes and DEGs were clustered according to the GO database to describe their functional attributes. All of the unigenes and DEGs were clustered into cellular components, molecular functions, and biological processes (Figure 1). Most DEGs in the “SDHB_AG vs. C_AG” comparison involved terms including “cytoplasm” (32 DEGs), “extracellular region” (27 DEGs), “metal ion binding” (22 DEGs), “integral component of membrane” (22 DEGs), and “ATP binding” (18 DEGs). In the “SDHB_T vs. C_T” comparison, most of the DEGs involved terms including “cytoplasm” (12 DEGs), “ATP binding” (10 DEGs), “cytosol” (8 DEGs), and “protein serine/threonine kinase activity” (7 DEGs).

In the KEGG pathway enrichment analysis, DEGs in the “SDHB_AG vs. C_AG” comparison were enriched including those related to hypertrophic cardiomyopathy (HCM), dilated cardiomyopathy (DCM), and the PI3K-Akt signaling pathway. DEGs in the “SDHB_T vs. C_T” comparison were enriched including those related to the citrate cycle (TCA cycle), oxidative phosphorylation, and the MAPK signaling pathway. The top 20 pathways with the largest number of DEGs are shown in Figure 2.

Notably, some DEGs were enriched in energy-related biological pathways, including the TCA cycle and the glycolysis/gluconeogenesis pathway (Figure 2). In the TCA cycle pathway, the *SDHB* gene was downregulated in both the “SDHB_AG vs. C_AG” comparison and the “SDHB_T vs. C_T” comparison, as a direct result of the RNAi treatment. Additionally, in the “SDHB_AG vs. C_AG” comparison, five genes were found to be downregulated in a set of relatively continuous biological processes related to the phosphorylated carbohydrate synthesis and transformation in glycolysis/gluconeogenesis pathways, including hexokinase, fructose-1,6-bisphosphatase I (*FBP*), triosephosphate isomerase (*TPI*), phosphoenolpyruvate carboxykinase-GTP (*PCK*-*GTP*), and multiple inositol-polyphosphate phosphatase (*MINPP1*) (Figure 3).

### 2.4. Male Sexual Development-Related DEGs

In the “SDHB_AG vs. C_AG” and “SDHB_T vs. C_T” comparisons, a total of nine male development-related genes were screened (Table 2). Among them, *Pro-resilin* is a DEG expressed during the embryonic development of male and female crustaceans, while *IAG* and the serine protease inhibitor 1 (*SERPINB1*) are related to hormone regulating processes, and the other six genes are related to spermiogenesis and sperm motility (Table 2). As shown in Table 2, after the knockdown of *SDHB* in the androgenic gland of *M. nipponense*, *Pro-resilin* and *IAG* were downregulated, while Protein mono-ADP-ribosyltransferase PAPR11 (*PARP11*) was upregulated. In the testes of *M. nipponense*, *DNAJC2* and C-type Lectin-1 (*CTL-1*) were upregulated, while *Pro-resilin*, Tyrosine-protein kinase Yes (*YES*), *Vigilin*, and sperm motility kinase Y-like (*SMOK-Y-like*) were upregulated.

### 2.5. Validation of DEGs by qRT-PCR

To validate the transcriptome results, six DEGs from each comparison that showed significantly different expression levels in the qRT-PCR analysis were selected at random. Positive numbers represent an upward trend, and negative numbers represent a downward trend. As shown in Figure 4, the expression patterns of the DEGs identified using qRT-PCR were generally similar to those obtained in the RNA-Seq analyses, although the relative expression levels were not completely consistent, thus proving that the transcriptome sequencing data were reliable.

## 3. Discussion

In this study, we attempted to identify male development-related genes and pathways regulated by *SDHB* in *M. nipponense*, based on the transcriptomes observed after *Mn-SDHB* knockdown using RNAi. This method can inhibit gene expression or translation through the short double-stranded RNA molecules in a cell’s cytoplasm [33], and has been widely used in gene function analysis in *M. nipponense*. Based on the RNAi treatment of *Mn-SDHB* dsRNA, a previous study found the inhibition of testis development and decrement in sperm in *M. nipponense*, indicating that *SDHB* may be involved in male sexual development in this species [12]. Because *Mn-SDHB* dsRNA was similarly employed in this study, based on both qRT-PCR and the study of the differential expression of the transcriptome, decreasing expression of *Mn-SDHB* was detected in the androgenic gland and testis in samples from day 7 following the *Mn-SDHB* dsRNA treatment, indicating the effectiveness of the dsRNA treatment. The qRT-PCR analysis of some representative DEGs suggested that the transcriptome sequencing and differential expression analysis were reliable.

In the androgenic gland of *M. nipponense*, five downregulated genes were detected in the glycolysis/gluconeogenesis pathway (which promotes the conversion of glucose into pyruvate, which releases free energy to form ATP, and pyruvate, which fuels the tricarboxylic acid cycle), as well as precursors for secondary metabolism, and amino acid and fatty acid biosynthesis. These DEGs are involved in a relatively continuous series of biological processes (Figure 3), indicating that the *Mn-SDHB* gene promotes the synthesis of phosphorylated carbohydrate and the transformations in these processes, and has an active role in the energy-related functions mentioned above. In addition, the glycolysis/gluconeogenesis pathway is involved in most biological processes, and has been shown to be an important pathway participating in male development in *M. nipponense*. In previous transcriptome studies in *M. nipponense*, the expression of many genes in the glycolysis/gluconeogenesis pathway was found to vary during the post-larval developmental stages 5 (PL5) to PL25, the sensitive periods for gonad development [34]. In addition, after ablation of the eyestalk, which is a key organ for secreting many hormones during male sexual development in crustaceans, the glycolysis/gluconeogenesis pathway is also one of the main sources of enrichment of the DEGs in the testes and androgenic gland of *M. nipponense* [35]. Moreover, the downregulated genes have also been shown to be associated with male sexual development in mammals. Phosphoenolpyruvate carboxykinase in the Leydig cells of prepubertal mouse testes have been shown to play an important role in steroidogenesis [36]. In a study of human disease, expression of the androgen receptor gene is abnormal in triosephosphate isomerase deficient patients, indicating that triosephosphate isomerase may affect the action of androgen [37]. There is also evidence that hexokinase, TPI, and MINPP1 are regulated by androgen [38,39,40]. Although the genes in the glycolysis/gluconeogenesis pathway have also been shown to play an important role in processes such as spermatogenesis or increasing sperm motility [41,42], this study detected no differences in the expression of these genes in the testes of *M. nipponense*, which might suggest that the different expression of *Mn-SDHB* does not affect these biological processes by regulating the glycolysis/gluconeogenesis pathway. According to previous in situ hybridization studies of the androgenic gland of *M. nipponense*, *SDHB* is expressed in the ejaculatory bulb surrounding the androgenic gland cells rather than in the androgenic gland cells themselves [43], indicating that *SDHB* might promote and support the formation of androgenic gland cells by influencing the glycolysis/gluconeogenesis process, and that it plays an essential role in maintaining the normal structure and function of the androgenic gland.

In this study, the differential expression of nine male sex-related genes was detected after *Mn-SDHB* knockdown. Among them is one gene that is differentially expressed during the embryonic development of male and female crustaceans. Pro-resilin is a type of resilin in a family of elastic proteins that includes elastin, as well as gluten, gliadin, abductin, and spider silks. *Pro-resilin* genes were found to be upregulated in males compared to females in the embryonic development of *M. rosenbergii* [44], a related species in the same genus as *M. nipponense*, indicating that it is important in the development of male prawn embryos. In this study, after *Mn-SDHB* knockdown, the *pro-resilin* gene was down-regulated in the androgenic gland and up-regulated in the testes of *M. nipponense*, showing that the effects of *Mn-SDHB* on pro-resilin vary in different male development-related tissues.

Two genes were related to the hormone regulatory process, *IAG* and *SERPINB1*. Coded by the *IAG* gene, insulin-like androgenic gland hormone is an insulin analogue of the insulin/insulin-like growth factor family, and has been shown to play essential roles in male sexual differentiation and development in crustacean species [27,45,46,47]. In the androgenic gland of *M. nipponense*, the expression of *Mn-IAG* was found to decrease after treatment with *Mn-SDHB* dsRNA, which is consistent with the results of previous qRT-PCR studies, and again points to the positive regulatory effect of *Mn-SDHB* on *Mn-IAG* in *M. nipponense*. SERPINB1 is an inhibitor of serine proteases, and resides mainly in the cytoplasm of neutrophils and monocytes, suppressing the enzymatic activities of serine proteases and preventing unwanted cellular damage during degranulation [48]. In rats, the *SERPINB1* gene was apparently down-regulated after a Sertoli cell-selective knockout of the androgen receptor, suggesting that it participates in tubular restructuring and cell junction dynamics processes in Sertoli cells, controlled in part by androgens [49]. *Mn-SERPINB1* was also downregulated in the androgenic gland of *M. nipponense* after *Mn-SDHB* knockdown.

Six genes were related to spermiogenesis and sperm motility. PAPR11 is a kind of ADP-ribosyltransferase poly polymerase, which plays a role in immune processes such as antiviral activity [50]. Deletion of the *PARP11* gene affects the shape of the cell nucleus and causes the formation of abnormally shaped fertilization-incompetent sperm, resulting in teratozoospermia and male infertility in mice. Therefore, it was predicted to have functional relevance for nuclear envelope stability and nuclear reorganization during spermiogenesis [51]. C-type lectins characteristically require calcium-related proteins to mediate essential cell functions through binding to carbohydrates. In *Ancylostoma ceylanicum*, the CTL-1 protein is a male gender-specific C-type lectin identified from sperm and soluble protein extracts and is important in hookworm reproductive physiology [52]. In studies on humans and mice, the testes/sperm galactosyl receptor was identified as a C-type lectin with possible roles in cell–cell interaction during spermatogenesis and sperm-zona pellucida binding at fertilization [53]. DNAJ chaperone was initially identified as a heat shock protein in *Escherichia coli*, and a prokaryotic homologue of the eukaryotic heat shock protein 70. It assists other proteins in their folding, transport, and assembly into complexes, which are necessary for testis development and spermatogenesis [54]. *DNAJC2* was described in bull sperm by Selvaraju in 2017 and is involved in stem cell differentiation and early embryonic development [55]. Tyrosine-protein kinase Yes Protein was shown to participate in the phosphorylation of tyrosine residues, whose truncated forms are related to spermiogenesis [55]. Vigilin, also known as high-density lipoprotein binding protein, is related to the processes of DNA damage repair [56], chromatin condensation, and gene silencing and according to a transcriptome study, its expression is the greatest in ovine sperm [55]. The *SMOK-Y-like* gene belongs to the sperm motility kinase family and is involved in sperm motility [57]. These male development-related genes were differentially expressed after the decreased expression of *Mn-SDHB*. However, only the *DNAJC2* and *CLT-1* found in the testes of *M. nipponense* were downregulated following *Mn-SDHB* treatment, indicating that the influence of *Mn-SDHB* on male development is complicated and requires further study.

## 4. Materials and Methods

### 4.1. Knockdown of the Expression of SDHB by RNAi in M. nipponense

The expression of *SDHB* of *M. nipponense* was knocked-down using RNAi. The specific RNAi primer for *SDHB* with a T7 promoter site was described in Appendix A in our previous study [12]. A Transcript Aid™ T7 High Yield Transcription kit (Fermentas Inc., Horsham, PA, USA) was used to synthesize *Mn-SDHB* dsRNA (*dsSDHB*) and GFP dsRNA (*dsGFP*), according to the manufacturer’s protocol. The *dsGFP* was used as a negative control [58]. A total of 300 mature male *M. nipponense* with body weights ranging from 3.26 to 4.76 g were collected. They were randomly divided into a *dsSDHB* treatment group (RNAi) and a *dsGFP* treatment group (control), and kept in separate 500 L tanks for 72 h prior to injection with either *dsSDHB* or *dsGFP*. The injected doses were 4 μg/g in both cases, consistent with the dose rates used in previous studies [59,60]. Androgenic gland and testis samples were collected from both the control and the RNAi groups on day 7 after *dsSDHB* and *dsGFP* injection, consistent with previous studies. qRT-PCR analysis revealed that the expression of *Mn-SDHB* decreased by over 90% in both the testes and androgenic glands of the *dsSDHB*-treated group, compared with the *dsGFP* group.

### 4.2. RNA Isolation, Library Construction, and Sequencing

The testes and androgenic glands of five *M. nipponense* from the *dsSDHB*-treated group (RNAi) and the *dsGFP*-treated group (control), were collected and individually pooled together to form a biological replicate, and three replicates were used for the transcriptome analysis.

Total RNA was extracted from each replicate using TRIzol reagent (Invitrogen, Waltham, MA, USA) according to the manufacturer’s protocol, followed by RNA purity and quantification evaluation using a Nano Drop 2000 spectrophotometer (Thermo Scientific, Waltham, MA, USA). RNA integrity was assessed using an Agilent 2100 Bioanalyzer (Agilent Technologies, Santa Clara, CA, USA). With the valid RNA samples (RIN ≥ 7, 28S:18S ≥ 0.7, >1 μg), libraries were then constructed using a VAHTS Universal V6 RNA-seq Library Prep Kit (Vazyme, Nanjing, China) according to the manufacturer’s instructions. The libraries were sequenced using an Illumina Novaseq 6000 platform (Illumina, San Diego, CA, USA) to generate the 150 bp paired-end reads.

### 4.3. Differential Gene Expression Analysis

Low-quality raw reads for each sample were removed from the data using fastp [61] with the default parameters. The clean reads obtained were mapped to the *M. nipponense* reference genome (Genbank access numbers: GCA_015110555.1 and GCA_015104395.1) using HISAT2 [62]. Gene expression was calculated using the FPKM method, where FPKM = cDNA fragments/mapped fragments (millions)/transcript length (kb), using HTSeq-count [63].

Differential expression analysis was performed using DESeq2 [64]. The false discovery rate (FDR) was calculated using the Benjamini–Hochberg correction method [65] with *q*-value < 0.05 and fold change >2 or <0.5 as the thresholds for significant DEGs.

Based on the hypergeometric distribution, with a threshold *p*-value < 0.05, the functional annotation and classification of DEGs were conducted according to the GO [66] and KEGG databases [67] to analyze the metabolic pathways of the enriched DEGs.

### 4.4. Quantitative Analysis

qRT-PCR was performed to evaluate the sequencing and data analysis and to validate the DEGs. Gonadal RNA was extracted (100 mg) using 1 mL TRIzol reagent (TaKaRa, Tokyo, Japan) and first-strand cDNA synthesis was performed using a Reverse Transcriptase M-MLV Kit (TaKaRa, Tokyo, Japan).

Genes were randomly selected from the DEGs for evaluation and their functional annotations, and the primers used are shown in Appendix A. The qRT-PCR was performed using an iCycler iQ5 real-time PCR system (Bio-Rad, Hercules, CA, USA), with eukaryotic translation initiation factor 5 A (*EIF*) as the reference gene, because of its stable expression in a variety of situations [68]. The reaction was amplified with 35 cycles of 94 °C for 30 s, 50 °C for 30 s, and 72 °C for 1 min, followed by 10 min incubation at 72 °C as the extension step. Each sample had three replicates while each reaction had three controls: nuclease-free water; primer-free water; and template-free water. The system recorded fluorescence curves and data automatically, and dissociation curves of the amplified products were recorded at the end of each PCR. The mRNA expression levels were determined using the 2^−ΔΔCT^ method [69].

### 4.5. Ethics Statement

The protocols of all experiments involving *M. nipponense* were approved by the Institutional Animal Care and Use Ethics Committee of the Freshwater Fisheries Research Center of the Chinese Academy of Fishery Sciences (Wuxi, China).

## 5. Conclusions

In this study, we knocked-down *Mn-SDHB* expression using RNAi and performed transcriptome studies of the androgenic gland and testes of *M. nipponense*. In the androgenic gland group treated with *dsSDHB*, five downregulated genes participating in phosphorylated carbohydrate synthesis and transformation were found in the glycolysis/gluconeogenesis pathway, which is important during the sexual development of male *M. nipponense*. Moreover, a total of nine male development-related genes were screened from the DEGs identified from the androgenic glands and testes of *M. nipponense*, demonstrating some of the regulatory effects of *Mn-SDHB* in the male development process of *M. nipponense*, and providing reference data for the further study of the mechanisms of male development in crustaceans in general.

## Figures and Tables

**Figure 1 ijms-24-13176-f001:**
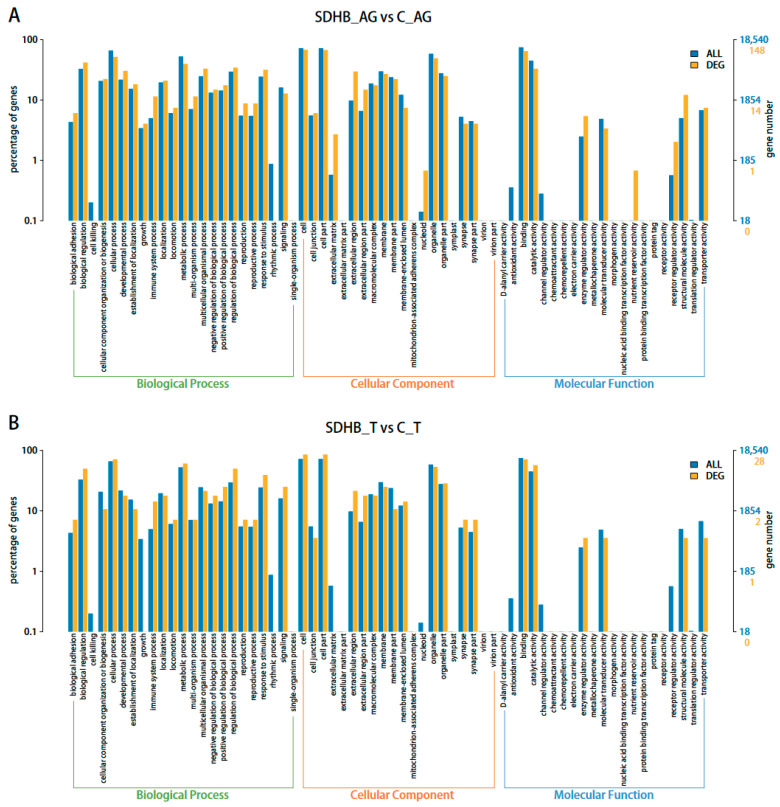
GO classification of the unigenes and DEGs discovered: (**A**) “SDHB_AG vs. C_AG”; (**B**) “SDHB_T vs. C_T”. The abscissa shows the second level terms in the three GO categories. The ordinates show the number of genes annotated to each term and the percentage of all genes.

**Figure 2 ijms-24-13176-f002:**
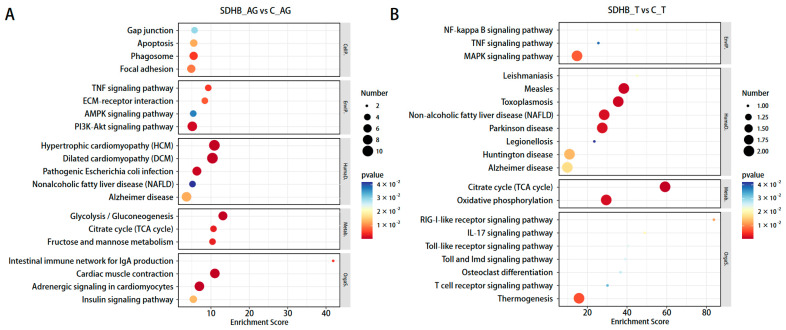
KEGG enrichment of DEGs: (**A**) “SDHB_AG vs. C_AG”; (**B**) “SDHB_T vs. C_T”. The size of the dots represents the number of genes. The color of each dot represents the *p*-value. The abscissa shows the enrichment score. The ordinate shows the number of genes annotated to each term and the percentage of all genes.

**Figure 3 ijms-24-13176-f003:**
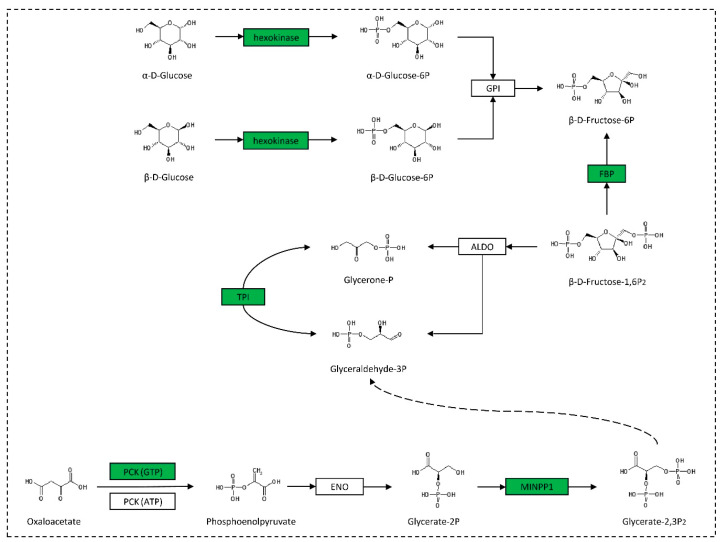
Schematic diagram of a partial glycolysis/gluconeogenesis pathway. The green boxes represent genes that are downregulated in the androgenic gland of *Macrobrachium nipponense*.

**Figure 4 ijms-24-13176-f004:**
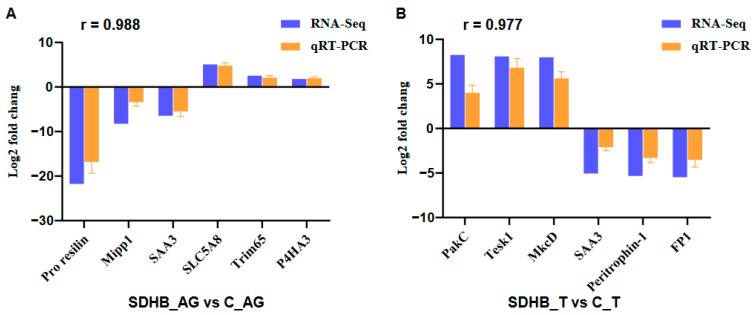
qRT-PCR validation of RNA-Seq data: (**A**) “SDHB_AG vs. C_AG”; (**B**) “SDHB_T vs. C_T”. r is the Pearson correlation coefficient between RNA-Seq group and qRT-PCR group.

**Table 1 ijms-24-13176-t001:** Statistics of the RNA-seq data in each of the three replicates.

Sample	Clean Bases (Gb)	Q30 (%)	GC (%)	Mapped Reads (%)	Unigene Number
C_AG1	7.15	92.49	45.89	91.58	16,623
C_AG2	6.89	92.47	45.34	91.23	16,731
C_AG3	6.94	92.48	45.14	91.76	17,035
C_T1	7.03	92.68	43.13	91.98	18,951
C_T2	6.76	92.78	43.11	91.28	19,421
C_T3	6.96	92.72	43.37	91.32	20,053
SDHB_AG1	6.98	92.58	45.7	92.19	16,623
SDHB_AG2	6.86	92.42	45.33	92.04	16,731
SDHB_AG3	7.14	92.87	45.32	91.31	17,035
SDHB_T1	7.3	92.61	43.59	91.62	18,951
SDHB_T2	7.13	92.65	43.59	91.48	19,421
SDHB_T3	7.2	92.84	43.48	92.22	20,053

**Table 2 ijms-24-13176-t002:** Male development-related genes screened from the differentially expressed genes.

Gene Name	Accession Number	Differential Expression	
		SDHB_AG vs. C_AG	SDHB_T vs. C_T
Pro-resilin	XM_043027950.1	down	up
IAG	XM_045267991.1	down	
SERPINB1	XM_045251131.1	down	
PARP11	XM_047641508.1	up	
DNAJC2	XM_042365393.1		down
CTL-1	XM_042349745.1		down
YES	XM_043000203.1		up
Vigilin	XM_027372514.1		up
SMOK-Y-like	XM_027369036.1		up

## Data Availability

The data presented in this study are available on request from the corresponding author for scientific purposes.

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
