# Peer review of "Identification of Male Sex-Related Genes Regulated by SDHB in Macrobrachium nipponense Based on Transcriptome Analysis after an RNAi Knockdown"

_ijms, 2023, doi:10.3390/ijms241713176_

Round 1
Reviewer 1 Report
In the present study, the authors identified several male development related genes in testis and androgenic gland by SDHB RNAi knockdown. In my opinion, there are some questions about the materials and methods and some results that need to be explained further for the main conclusions of the manuscript. And the manuscript requires major language editing.
1. In the Abstract, the authors should state the purpose and significance of this research.
2. In Figure4, the authors should add the significant / the P value.
3. In M&M, How the authors choose the 4 μg/g? is there a pre-experiment?
4. Also in M&M, in my opinion, the authors should add the development stage of the M. nipponense for it is vital for exploring the effect of the SDHB. In addition, the authors should add a figure of the histology of the testis.
5. There are specific points about the writing, Line 35, Line 182, Line 203. Please check all the manuscript.
The manuscript requires major language editing.
Author Response
In the present study, the authors identified several male development related genes in testis and androgenic gland by SDHB RNAi knockdown. In my opinion, there are some questions about the materials and methods and some results that need to be explained further for the main conclusions of the manuscript. And the manuscript requires major language editing.
- In the Abstract, the authors should state the purpose and significance of this research.
The purpose and significance of this research have been supplied in the Abstract, “This study knocked-down the Mn-SDHB genes in M. nipponense using RNAi. A transcrip-tome analysis of the androgenic gland and testes was then performed to discover the male sex-related genes regulated by SDHB and investigate the mechanism of male sexual devel-opment in this species.”.
- In Figure4, the authors should add the significant / the P value.
To show the correlation, we calculated the Pearson correlation coefficient and add it to Figure 4. In addition, based on the standard error, the error bars for qRT-PCR have been added.
- In M&M, How the authors choose the 4 μg/g? is there a pre-experiment?
Previous study revealed that does of 4 μg/g had a significant interference effect. The result was first presented in a Chinese literature, and our subsequent studies, including this one, used the same dose.
Jiang, F. W., Fu, H. T., Qiao, H., Zhang, W. Y., Jiang, S. F., Xiong, Y. W., ... & Jin, S. B. (2014). The RNA interference regularity of transformer-2 gene of oriental river prawn Macrobrachium nipponense. Chin. Agric. Sci. Bull, 30, 32-37.
- Also in M&M, in my opinion, the authors should add the development stage of the nipponense for it is vital for exploring the effect of the SDHB. In addition, the authors should add a figure of the histology of the testis.
We used mature male M. nipponense about 4 months after hatching as the samples. Since In situ hybridization analysis of Mn-SDHB gene in testis and androgenic gland from reproductive season of M. nipponense have been published, we didn’t add a figure of the testis. The hematoxylin and eosin staining can be found in the corresponding document.
Jin, S., Hu, Y., Fu, H., Jiang, S., Xiong, Y., Qiao, H., ... & Wu, Y. (2021). Identification and characterization of the succinate dehydrogenase complex iron sulfur subunit B gene in the Oriental River Prawn, Macrobrachium nipponense. Frontiers in Genetics, 12, 698318.
- There are specific points about the writing, Line 35, Line 182, Line 203. Please check all the manuscript.
The manuscript has been revised by a language editing service, and these sentences have been rewritten.
Reviewer 2 Report
The manuscript ijms-2560036 entitled “Identification of male development related genes regulated by SDHB of Macrobrachium nipponense based on the transcriptome after an RNAi knockdown” provides comprehensive scientific results about the role of SDHB in the sexual development of male Oriental river prawn (Macrobrachium nipponense). The authors used RNAi to knock down the gene. Later they applied the mRNA-seq to identify the molecular pathway and genes that participate in this mechanism. In general, the topic is important for the readers of IJMS and also for aquaculture scientists. This research has some innovations because the authors find and reveal the role of a novel sex-related gene in crustaceans.
I would like to recommend the acceptance of this manuscript after a minor revision. However, the English language needs a major and critical revision.
General comments:
There are several errors in the English language and style. I just highlighted some of them. The manuscript must undergo revision by a language editing service or a native speaker.
My specific comments are as below:
Title: “Identification of male sex-related genes... ”
Abstract
Line 12-15: The sentence needs revision
Line 15-18: The sentence needs revisions there are so many “,”. In general, the sentence is not written very well.
Line 16: “transcriptome analysis”
Line 16: Change “with” to “of”.
Line 19: hypertrophic cardiomyopathy pathway
Line 20: citrate cycle pathway
Line 21: in a series of biological processes associated with phosphorylated carbohydrates synthesis and…”
Line 23: a total of nine male sex-related genes including Pro-resilin, insulin-like androgenic gland hormone, Protein mono-ADP-ribosyltransferase PAPR11, DNAJC2, C-type Lectin 1, Tyrosine-protein kinase Yes, Vigilin and sperm motility kinase Y-like were identified, revealing the regulatory…”
Line 24-27: Why the name of some genes started with capital letters and why some are written in lowercase. Better to be unique.
Line 28: male sexual development
1. Introduction
Line 33: “in in”. Remove one of the “in”
Line 34: “regions of China”
Line 36-37: Which species?
Line 40: What means “carry out the reproduction mechanism”? Do you mean “understanding of the reproduction mechanism”?
Line 52: plays
Line 60-61: Check the line space
Line 63: male sexual differentiation
Line 64: existing
Line 64-65: This sentence is already written before.
Line 67: What means “focused”?
Line 68: male sexual differentiation
Line 69: You can add some basic information about the relation mechanism of IAG and its relation with other sex-related genes to line 69. There are several review papers about this mechanism, I provided some of them for you as a reference.
The regulation mechanism of IAG and sexual development in crustaceans is very complicated and relies on the expression of multiple genes. Recent studies showed that several sex-related genes including CFSH, Vtg, Wnt4, Dmrt gene family, Sox gene family, cell cycle gene family, and Fem-1 participate in the regulation of IAG, either directly or indirectly. Nevertheless, further investigations are required to explore the interrelationship between sex-related genes and their association with IAG.
A sex-reversing factor: insulin-like androgenic gland hormone in decapods, Reviews in aquaculture, 2021
The regulatory mechanism of sexual development in decapod crustaceans. Frontiers in Marine Science, 8, p.679687. 2021
Sex determination and developmental mechanism of crustaceans and shellfish-volume II. Frontiers in Endocrinology, 14, p.453. 2023
The significant sex-biased expression pattern of Sp-Wnt4 provides novel insights into the ovarian development of mud crab (Scylla paramamosain). International Journal of Biological Macromolecules, 183, pp.490-501. 2021
Line 69: “The IAG knockdown had a significant…”
2. Results
Line 82-86: Please divide the sentence into two separate sentences.
Line 102-107: Must be transferred to the materials and methods section.
Line 175: why there is no error bar, especially for qRT-PCR data. Did you use only one sample?
3. Discussion
Line 178: Remove “expression”
Line 179: Change “with” to “using”
Line 181-190: There are a lot of “,” in these sentences. The writing style must be revised.
Line 182: “previous”. “p” must be in lowercase.
Line 203: “after”. “a” must be in lowercase.
Line 207: “male sexual development”.
Line 211: “There are…”. NOTE: “And” cannot be used to start a sentence.
Line 218: “in situ” must be written in italic format.
Line 218-223: There are a lot of “,” in these sentences. The writing style must be revised. Better to divide the sentences into smaller sentences.
Line 224: “In this study,”
Line 224: male sex-related genes
Line 235: Change “are” to “were”
Line 240: Change “of” to “with”
Line 242: Remove “again”
Line 250: Change “are” to “were”
Line 260: Change “with” to “on”
Line 263: “Escherichia coli” must be written in italic format.
4. Materials and Methods
Line 292: why after 7 days, for example, why not after 48h?
Line 293: “expression was”
Line 304: What was the cafeteria for considering the RNA sample to be suitable for sequencing? OD260/280=???, OD260/230≥????, RIN≥???, 28S:18S≥???, >??μg
Line 321: What was the P-value for GO and KEGG enrichment analysis?
There are several errors in the English language and style. I just highlighted some of them. The manuscript must undergo revision by a language editing service or a native speaker.
Author Response
The manuscript ijms-2560036 entitled “Identification of male development related genes regulated by SDHB of Macrobrachium nipponense based on the transcriptome after an RNAi knockdown” provides comprehensive scientific results about the role of SDHB in the sexual development of male Oriental river prawn (Macrobrachium nipponense). The authors used RNAi to knock down the gene. Later they applied the mRNA-seq to identify the molecular pathway and genes that participate in this mechanism. In general, the topic is important for the readers of IJMS and also for aquaculture scientists. This research has some innovations because the authors find and reveal the role of a novel sex-related gene in crustaceans.
I would like to recommend the acceptance of this manuscript after a minor revision. However, the English language needs a major and critical revision.
General comments:
There are several errors in the English language and style. I just highlighted some of them. The manuscript must undergo revision by a language editing service or a native speaker.
Thank you for your comments. The manuscript has been revised by a language editing service.
My specific comments are as below:
Title: “Identification of male sex-related genes... ”
It has been revised.
Abstract
Line 12-15: The sentence needs revision
The sentence has been revised.
Line 15-18: The sentence needs revisions there are so many “,”. In general, the sentence is not written very well.
The sentence has been rewritten.
Line 16: “transcriptome analysis”
It has been revised.
Line 16: Change “with” to “of”.
It has been changed.
Line 19: hypertrophic cardiomyopathy pathway
It has been revised.
Line 20: citrate cycle pathway
It has been revised.
Line 21: in a series of biological processes associated with phosphorylated carbohydrates synthesis and…”
It has been revised.
Line 23: a total of nine male sex-related genes including Pro-resilin, insulin-like androgenic gland hormone, Protein mono-ADP-ribosyltransferase PAPR11, DNAJC2, C-type Lectin 1, Tyrosine-protein kinase Yes, Vigilin and sperm motility kinase Y-like were identified, revealing the regulatory…”
It has been revised.
Line 24-27: Why the name of some genes started with capital letters and why some are written in lowercase. Better to be unique.
All genes mentioned start with capital letters now.
Line 28: male sexual development
It has been revised.
- Introduction
Line 33: “in in”. Remove one of the “in”
It has been revised.
Line 34: “regions of China”
It has been revised.
Line 36-37: Which species?
It is M. nipponense. The sentence has been revised.
Line 40: What means “carry out the reproduction mechanism”? Do you mean “understanding of the reproduction mechanism”?
Yes, it has been changed to “understand the reproduction mechanism”.
Line 52: plays
It has been revised.
Line 60-61: Check the line space
It has been revised.
Line 63: male sexual differentiation
It has been revised.
Line 64: existing
It has been revised.
Line 64-65: This sentence is already written before.
It has been removed.
Line 67: What means “focused”?
It means that IAG is specifically expressed in the androgenic gland. The sentence has been revised.
Line 68: male sexual differentiation
It has been revised.
Line 69: You can add some basic information about the relation mechanism of IAG and its relation with other sex-related genes to line 69. There are several review papers about this mechanism, I provided some of them for you as a reference.
The regulation mechanism of IAG and sexual development in crustaceans is very complicated and relies on the expression of multiple genes. Recent studies showed that several sex-related genes including CFSH, Vtg, Wnt4, Dmrt gene family, Sox gene family, cell cycle gene family, and Fem-1 participate in the regulation of IAG, either directly or indirectly. Nevertheless, further investigations are required to explore the interrelationship between sex-related genes and their association with IAG.
A sex-reversing factor: insulin-like androgenic gland hormone in decapods, Reviews in aquaculture, 2021
The regulatory mechanism of sexual development in decapod crustaceans. Frontiers in Marine Science, 8, p.679687. 2021
Sex determination and developmental mechanism of crustaceans and shellfish-volume II. Frontiers in Endocrinology, 14, p.453. 2023
The significant sex-biased expression pattern of Sp-Wnt4 provides novel insights into the ovarian development of mud crab (Scylla paramamosain). International Journal of Biological Macromolecules, 183, pp.490-501. 2021
We really appreciate your advice. These have been added to the manuscript.
Line 69: “The IAG knockdown had a significant…”
It has been revised.
- Results
Line 82-86: Please divide the sentence into two separate sentences.
The sentence has been rewritten.
Line 102-107: Must be transferred to the materials and methods section.
It has been revised.
Line 175: why there is no error bar, especially for qRT-PCR data. Did you use only one sample?
Error bars for qRT-PCR were added.
We used three replicates for each sample. Based on the standard error, the error bars for qRT-PCR have been added.
- Discussion
Line 178: Remove “expression”
It has been revised.
Line 179: Change “with” to “using”
It has been revised.
Line 181-190: There are a lot of “,” in these sentences. The writing style must be revised.
The sentences have been revised.
Line 182: “previous”. “p” must be in lowercase.
It has been revised.
Line 203: “after”. “a” must be in lowercase.
It has been revised.
Line 207: “male sexual development”.
It has been revised.
Line 211: “There are…”. NOTE: “And” cannot be used to start a sentence.
Thank you for reminding us. It has been revised.
Line 218: “in situ” must be written in italic format.
It has been revised.
Line 218-223: There are a lot of “,” in these sentences. The writing style must be revised. Better to divide the sentences into smaller sentences.
It has been revised.
Line 224: “In this study,”
It has been revised.
Line 224: male sex-related genes
It has been revised.
Line 235: Change “are” to “were”
It has been changed.
Line 240: Change “of” to “with”
It has been changed.
Line 242: Remove “again”
It has been removed.
Line 250: Change “are” to “were”
It has been changed.
Line 260: Change “with” to “on”
It has been changed.
Line 263: “Escherichia coli” must be written in italic format.
It has been revised.
- Materials and Methods
Line 292: why after 7 days, for example, why not after 48h?
Previous study revealed that gene expression changes are most pronounced at 7 days after the RNAi treatment. The result was first presented in a Chinese literature, and our subsequent studies, including this one, used the same time.
Jiang, F. W., Fu, H. T., Qiao, H., Zhang, W. Y., Jiang, S. F., Xiong, Y. W., ... & Jin, S. B. (2014). The RNA interference regularity of transformer-2 gene of oriental river prawn Macrobrachium nipponense. Chin. Agric. Sci. Bull, 30, 32-37.
Line 293: “expression was”
It has been revised.
Line 304: What was the cafeteria for considering the RNA sample to be suitable for sequencing? OD260/280=???, OD260/230≥????, RIN≥???, 28S:18S≥???, >??μg
The RNA samples for sequencing must meet the following requirements, RIN ≥ 7, 28S:18S ≥ 0.7, > 1 μg. The information has been added to the manuscript.
Line 321: What was the P-value for GO and KEGG enrichment analysis?
The p-value < 0.5 is the threshold for the GO and KEGG enrichment analysis. The information has been added to the manuscript.
Round 2
Reviewer 1 Report
The authors have revised all the comments. And the manuscript has met the requirements for publication.
English language required more editing.